

# Impact of public sentiments on the transmission of COVID-19 across a geographical gradient

Folashade B. Agusto[1], Eric Numfor[2], Karthik Srinivasan[1],
Enahoro A. Iboi[3], Alexander Fulk[1], Jarron M. Saint Onge[1,4] and
A. Townsend Peterson[1]

[1] University of Kansas, Lawrence, Kansas, United States
[2] Augusta University, Augusta, Georgia, United States
[3] Spelman College, Atlanta, Georgia, United States
[4] University of Kansas Medical Center, Kansas City, Kansas, United States

Corresponding author
Folashade B. Agusto,
fbagusto@gmail.com

## ABSTRACT

COVID-19 is a respiratory disease caused by a recently discovered, novel coronavirus, SARS-COV-2. The disease has led to over 81 million confirmed cases of COVID-19, with close to two million deaths. In the current social climate, the risk of COVID-19 infection is driven by individual and public perception of risk and sentiments. A number of factors influences public perception, including an individual's belief system, prior knowledge about a disease and information about a disease. In this article, we develop a model for COVID-19 using a system of ordinary differential equations following the natural history of the infection. The model uniquely incorporates social behavioral aspects such as quarantine and quarantine violation. The model is further driven by people's sentiments (positive and negative) which accounts for the influence of disinformation. People's sentiments were obtained by parsing through and analyzing COVID-19 related tweets from Twitter, a social media platform across six countries. Our results show that our model incorporating public sentiments is able to capture the trend in the trajectory of the epidemic curve of the reported cases. Furthermore, our results show that positive public sentiments reduce disease burden in the community. Our results also show that quarantine violation and early discharge of the infected population amplifies the disease burden on the community. Hence, it is important to account for public sentiment and individual social behavior in epidemic models developed to study diseases like COVID-19.

## INTRODUCTION

COVID-19 is caused by a coronavirus called the severe acute respiratory syndrome coronavirus-2 (SARS-CoV-2). Coronaviruses are a large family of viruses that are common in humans and many different species of animals, including camels, cattle, cats, and bats (*Centers for Disease Control and Prevention, 2020a*; *WHO, 2020d*). This virus was discovered in Wuhan China, in 2019, and has since been declared a pandemic by the World Health Organization (WHO). As of December 31, 2020, there were over 81 million

confirmed cases of COVID-19, with close to two million deaths globally (*Dong, Du & Gardner, 2020*; *WHO, 2020e*). According to COVID-19 data obtained from the Johns Hopkins University Center for Systems Science and Engineering COVID-19 Dashboard (*Dong, Du & Gardner, 2020*), the United States had the highest cumulative number of cases with nearly 20 million confirmed cases and over 340,000 reported deaths. Brazil has the next highest deaths with with over 190,000 deaths and over seven million cases. Cases are also rising across Africa; South Africa has the highest number of confirmed cases: over one million cases with over 28,000 deaths. These statistics clearly show that humans were not efficient in curtailing the spread of the novel virus.

The virus can be transmitted from person-to-person *via* direct contact with respiratory droplets or by touching contaminated surfaces and objects containing the virus; the virus can live on contaminated surfaces and objects (*Centers for Disease Prevention and Control, 2020f*; *Petersen et al., 2020*; *Yung et al., 2020*). The incubation period for those exposed to COVID-19 varies from 2 to 14 days after exposure to the virus (*Centers for Disease Control and Prevention, 2020a*, *2020c*, *2020e*). However, onset of symptoms is often seen earlier in people with pre-existing health conditions and compromised immune systems. Reports indicate that patients with mild symptoms take a week or more to recover, while cases that are severe may gradually progress to respiratory failure, which may lead to death. More serious complications from COVID-19 illness leading to death are more common in middle-aged and elderly patients who have severe underlying medical conditions like heart or lung disease, diabetes, or cirrhosis (*Adeniyi et al., 2020*). There is a wide range of symptoms observed in patients with COVID-19, including fever, shortness of breath, dry cough, headaches, nausea, sore throat, chest pain, loss of taste or smell, diarrhea, and severe fatigue (*Centers for Disease Prevention and Control, 2020e*).

Recently, therapeutics such as Remdesivir have been approved for treatment of hospitalized individuals; vaccines are being approved but are not yet wildly available and only essential workers and the elderly are currently being vaccinated (*US Food and Drug Administration, 2020a*, *2020b*, *2020c*, *2020d*). As such, non-pharmaceutical interventions such as social distancing, school and event closings, travel bans, community lockdowns, contact tracing, quarantine of confirmed cases, and the use of face masks in public are continually being used as mitigation efforts against the virus transmission. Social distancing guidelines as suggested by the Centers for Disease Control and Prevention (CDC) (*Centers for Disease Prevention and Control, 2020d*) and the World Health Organization (*WHO, 2020f*) state that individuals outside their homes should be six feet apart from other people and must wear a face-mask at all times. The use of face masks in public by members of the general population has historically been a common practice to combat the spread of respiratory diseases, dating back to at least the 1918 H1N1 pandemic of influenza (*Bootsma & Ferguson, 2007*). The guidelines further recommend that people frequently wash their hands for at least 20 s, even in their homes, as research has shown that soap kills the virus and reduces one's chance of getting infected (*Centers for Disease Prevention and Control, 2020d*). Infected individuals and suspected cases are quarantined or advised to self-isolate. However, little is known about best management strategies for limiting further transmission and spread. Furthermore, the success of these preventive
measures depends on voluntary compliance by the population (*Agusto et al., 2022*), and may depend in part to perceptions and interpretations of risk.

The response of individuals in the community to the threat of an infectious disease is dependent on their perception of risk, which can be swayed by public and private information disseminated through diverse media. Many individuals use social media platforms like Twitter, Facebook, and the internet more generally to share social and health information, and many have used these platforms to also spread misinformation and conspiracy theories. Many health-related organizations also use these platforms to send information to mitigate the spread of contagious diseases (like the flu) by educating users on the effectiveness of regular hand-washing, use of face masks, social distancing, and raising awareness about vaccines (*Philipose, 2020*). For instance, in the past decade, the Centers for Disease Control made use of Twitter in disseminating information on the prevention of flu to help curb the spread of H1N1 influenza in 2009 (*Philipose, 2020*). Media reporting is important in the perception, management and even creation of crises (*Marino et al., 2009*; *Tchuenche et al., 2011*). Information provided to the public through the media changes human behavior and the population adopts the precautionary measures like the use of face masks for influenza (*Jenco, 2020*), vaccination (*Aminiel, Kajunguri & Mpolya, 2015*; *Buonomo, d'Onofrio & Lacitignola, 2008*), and voluntary quarantine (*Hethcote, Ma & Shengbing, 2002*). Thus, the role of media coverage and social media responses on disease outbreaks is crucial and should be given prominence in the study of disease dynamics.

Numerous mathematical models have been used to gain insight into the effect of media and behavioural change on COVID-19 transmission dynamics. A SEIQR-type compartmental model was developed in (*Feng et al., 2020*) to assess the impact of media coverage and quarantine on the COVID-19 infections in the UK. The study showed that stringent containment strategies should be adopted in the UK in order to effectively curtail the spread of the disease. *Aleta et al. (2020)* used a stochastic model to understand the impact of testing, contact tracing and household quarantine on second waves of COVID-19 in the Boston metropolitan area. Their result showed that a response system based on enhanced testing and contact tracing can have a major role in relaxing social-distancing interventions in the absence of herd immunity against COVID-19. *Eikenberry et al. (2020)* developed a compartmental model to assess the community-wide impact of mask use by the general asymptomatic public. The study showed that broad adoption of even relatively ineffective face masks could reduce community transmission of COVID-19 and decrease peak hospitalizations and deaths. A mathematical model was developed in *Iboi et al. (2021)* to assess the impact of a public health education program on the coronavirus outbreak in the United States. Their result suggests the need to obey public health measures as loss of willingness would increase the cumulative and daily mortality in the United States.

Our objective in this study is to gain insight into the contribution of human behavior and public sentiment to the disease spread and not to make explicit epidemiological predictions and forecasting about the disease outbreak. Here, we use tweets as a source of public sentiment data and analyze their average parity (*i.e.*, negativeness and positiveness)
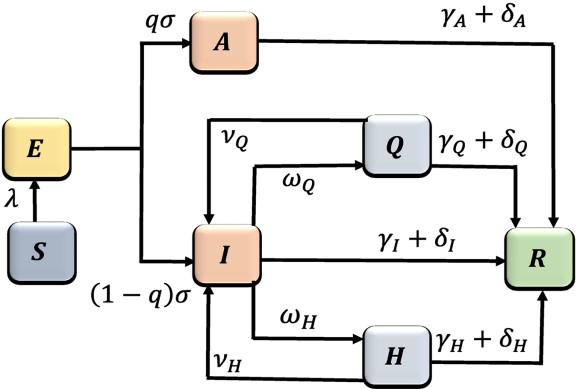

**Figure 1 Flow diagram of the COVID-19 model (1),** where $\lambda = \frac{\beta[I(t)+\eta_A A(t)+\eta_Q Q(t)+\eta_H H(t)]S(t)}{N(t)}$.

across six countries experiencing variations in disease response and spread (Australia, Brazil, Italy, South Africa, United Kingdom, and United States) during January to May, 2020 time period. Tweets are short messages limited to 240 characters that are posted in real time by users of the Twitter social media platform.

The remainder of the work in this article is organized as follows. In Section 2, we formulate our baseline COVID-19 model with human behavior, compute the basic reproduction number of the model, fit COVID-19 data to the model, and estimate parameters of the model. In Section 3, we carry out sensitivity analysis of the basic reproduction number with respect to each parameter, and sentiment analysis is carried out in Section 4. In Section 5, we incorporate sentiment effects in our basic model, and present results in Section 6. Our discussion and conclusions are presented in Section 7.

## BASELINE COVID-19 MODEL

To formulate the COVID-19 model with human behavior where some individuals violate quarantine rules, we followed the natural history of the infection (*Picheta, 2020*; *Wilson & Kluger, 2020*) and segment the population according to their disease status as susceptible $(S(t))$, exposed $(E(t))$, asymptomatic $(A(t))$, symptomatic $(I(t))$, quarantined $(Q(t))$, hospitalized $(H(t))$, and removed $(R(t))$. The equations of the mathematical model are given in Eq. (1). A flow diagram depicting the transition from one state to the other as the disease progresses through the population is shown in Fig. 1, and the associated state variables and parameters are described in Table 1.

The population of susceptible $(S(t))$ is decreased by infection at the rate $\frac{\beta[I(t)+\eta_A A(t)+\eta_Q Q(t)+\eta_H H(t)]S(t)}{N(t)}$, where $\beta$ is the infection rate; we assume that $\eta_A, \eta_Q, \eta_H < 1$, meaning that the asymptomatic, quarantined, and hospitalized are not as infectious as the symptomatic individuals. Once infected, the susceptible move into the exposed class $(E(t))$ and a portion of the exposed population develops clinical symptoms of the disease at the rate $(1-q)\sigma$ and move into the infectious class $(I(t))$, while the remaining proportion shows no symptoms and moves into the asymptomatic class $(A(t))$ at the rate $q\sigma$. The symptomatic individuals either are quarantined at the rate $\omega_Q$ or are hospitalized at

| Table 1 Description of the variables and parameters of the COVID-19 model (1). | |
|---|---|
| **Variable** | **Description** |
| $S(t)$ | Number of susceptible individuals |
| $E(t)$ | Number of exposed individuals |
| $A(t)$ | Number of asymptomatic infectious individuals |
| $I(t)$ | Number of symptomatic infectious individuals |
| $Q(t)$ | Number of quarantined individuals |
| $H(t)$ | Number of hospitalized individuals |
| $R(t)$ | Number of removed individuals |
| **Parameter** | **Description** |
| $\beta$ | Infection rate |
| $\eta_A$ | Infection modification parameter for the asymptomatic infection rate |
| $\eta_Q$ | Infection modification parameter for the quarantined infection rate |
| $q$ | Proportion of exposed developing asymtomatic infections |
| $\sigma$ | Disease progression rate from the exposed to either asymptomatic or infectious |
| $\gamma_I$ | Recovery rate of infectious |
| $\gamma_A$ | Recovery rate of asymptomatic |
| $\gamma_Q$ | Recovery rate of quarantined |
| $\gamma_H$ | Recovery rate of those hospitalized |
| $\omega_Q$ | Quarantine rate |
| $\omega_H$ | Hospitalization rate |
| $v_Q$ | Quarantine violation rate |
| $v_H$ | Hospital discharge rate |
| $\delta_Q$ | Death rate of quarantined |
| $\delta_H$ | Death rate of hospitalized |
| $\delta_I$ | Death rate of infectious |
| $\delta_A$ | Death rate of asymptomatic |

the rate $\omega_H$. There have been several reports of people flouting mandatory quarantine rules (*Choi, 2020*; *Crane, 2020*; *Frias, 2020*; *Neuman, 2020*), so we assume that individuals in quarantine violate the quarantine rules/laws at the rate $v_Q$. The alarming rate at which the disease spreads and people require hospitalization, hospitals may become overwhelmed and could run out of beds, respirators, ventilators, and ICUs (*Starleaf Riker & Chasnoff, 2020*). Furthermore, some hospitals are reserving beds for critically ill COVID-19 patients and discharging to nursing homes those with less severe illness (*Baker & Fink, 2020*; *Graham, 2020*). Thus, we assume that, due to limitations in hospital beds, respirators, ventilators, and ICUs, some hospitalized leave the hospitals at the rate $v_H$. We also assume that once an individual is infected they remain immune to the virus. The removed class ($R(t)$) tracks either the recovered at the rates $\gamma_I, \gamma_A, \gamma_Q, \gamma_H$ or those that have died due to COVID-19 at the rates $\delta_I, \delta_A, \delta_Q, \delta_H$, from the symptomatic, asymptomatic, quarantined, and hospitalized classes, respectively. The equations of the mathematical model are given in Eq. (1).

$$\frac{dS}{dt} = -\frac{\beta[I(t) + \eta_A A(t) + \eta_Q Q(t) + \eta_H H(t)]S(t)}{N(t)}$$

$$\frac{dE}{dt} = \frac{\beta[I(t) + \eta_A A(t) + \eta_Q Q(t) + \eta_H H(t)]S(t)}{N(t)} - \sigma E(t)$$

$$\frac{dA}{dt} = q\sigma E(t) - \gamma_A A(t) - \delta_A A(t)$$

$$\frac{dI}{dt} = (1-q)\sigma E(t) + v_Q Q(t) + v_H H(t) - \omega_Q I(t) - \omega_H I(t) - \gamma_I I(t) - \delta_I I(t)$$

$$\frac{dQ}{dt} = \omega_Q I(t) - \gamma_Q Q(t) - v_Q Q(t) - \delta_Q Q(t)$$

$$\frac{dH}{dt} = \omega_H I(t) - \gamma_H H(t) - v_H H(t) - \delta_H H(t)$$

$$\frac{dR}{dt} = (\gamma_I + \delta_I)I(t) + (\gamma_A + \delta_A)A(t) + (\gamma_Q + \delta_Q)Q(t) + (\gamma_H + \delta_H)H(t)$$

$$(1)$$

where $N(t) = S(t) + E(t) + I(t) + A(t) + Q(t) + H(t) + R(t)$.

The associated reproduction number (*Diekmann, Heesterbeek & Metz, 1990*; *van den Driessche & Watmough, 2002*) of the COVID-19 model (1), denoted by $\mathcal{R}_0$, is given by

$$\mathcal{R}_0 = \mathcal{R}_{0I} + \mathcal{R}_{0A},$$
$$= \frac{(1-q)\beta(k_3 k_4 + \eta_Q \omega_Q k_4 + \eta_H \omega_H k_3)}{(k_1 k_3 k_4 - \omega_Q v_Q k_4 - \omega_H v_H k_3)} + \frac{q\beta\eta_A}{k_2}. \quad (2)$$

where, $k_1 = \gamma_I + \omega_Q + \omega_H + \delta_I, k_2 = \gamma_A + \delta_A, k_3 = v_Q + \gamma_Q + \delta_Q, k_4 = v_H + \gamma_H + \delta_H$.

The expression $\mathcal{R}_{0I}$ represents the contribution of the symptomatic infectious individuals to the reproduction, and the expression $\mathcal{R}_{0A}$ represents the contribution to reproduction number due to the asymptomatic individual. The reproduction number, $\mathcal{R}_0$, is the average number of secondary infectious produced when a single infected individual is introduced into a completely susceptible population (*Diekmann, Heesterbeek & Metz, 1990*; *van den Driessche & Watmough, 2002*). Hence, COVID-19 can be effectively controlled in the population if the reproduction number can be reduced to (and maintained at) a value less than unity (*i.e.*, $\mathcal{R}_0 < 1$).

## Data fitting and parameter estimation

Some of the parameters of the model (1) were obtained from literature, while others were obtained by fitting the model to the observed cumulative case data for each of the six countries (Australia, Brazil, Italy, South Africa, United Kingdom, and United States) during January–June, 2020 (see Tables A1 and A2 for initial conditions and estimated parameters). The cumulative case data from the respective first index case of each of the countries to June 19, 2020 were obtained from the John Hopkins' center for systems science and engineering COVID-19 Dashboard (*Dong, Du & Gardner, 2020*). During this time period, these countries instituted lockdowns in March 2020 as a means to control and contain the disease. Italy instituted a lockdown on March 9, Brazil March 17, US March 19, Australia March 23, UK March 23, and South Africa March 26 (*WHO, 2020a*). Thus, the model was fitted to the two different time periods, the first period is the time before each of

the countries instituted lockdown measures to curtail the virus and the second period is after lockdown was in place. We obtained two different sets of parameters for some parameters in each of the time periods; others remained the same, for instance the death rate, the disease progression rate, the proportion of asymptomatic did not change over this time period.

We estimate the remaining five parameters, $\beta, \omega_Q, \omega_H, v_Q$ and $v_H$, of the model using the MultiStart algorithm with the *fmincon* function in MATLAB's optimization toolbox (*Burton et al., 2021*; *Che, Kang & Yakubu, 2020*; *Che et al., 2021*; *Edholm et al., 2019*; *Edholm et al., 2022*; *Loria, 2018*). The fitting was implemented by formulating a least-squares optimization problem with the aim of minimizing the difference between the cumulative cases in each of the six countries and our model's output. The objective function minimized is given as

$$J = \frac{\|YC - YC^*\|_2}{\|YC^*\|_2}, \tag{3}$$

where the vector $YC$ contains the cumulative number of infections obtained from the model and the vector $YC^*$ contains the corresponding values from the data.

Our parameter estimation simulations begin on dates cases were reported in each of the six countries and take daily time steps until the date our data ends, which is June 19, 2020. The values of the initial conditions used for the fitting are given in Table A1. Given a starting point for our objective function $J$, the fmincon algorithm outputs a local minimum on the surface of $J$. To help find the global minimum, MultiStart allows us to exhaustively test different starting values throughout our bounded range. We used different starting points, each of which converged to a unique local minimum on the surface of $J$. Considering the United Kingdom, the smallest objective function value obtained before and after the lockdown are $J_0 = 0.17$ and $J_1 = 0.02$, respectively.

Figure 2A shows the fitting of the observed cumulative cases for the United Kingdom before lockdown was put in place. The estimated values of the fitted parameters are tabulated in Table 2. The fitting for after lockdown for UK is depicted in Fig. 2B and the estimated parameter values used as well as parameters for the other countries are given in Table A2.

The numerical value of the reproduction number $\mathcal{R}_0$ for United Kingdom before the country's lockdown was put in place is estimated using the parameter values tabulated in Table 2. Consequently, using these parameter estimates, we obtain the value of $\mathcal{R}_0$ for the COVID-19 outbreak in United Kingdom before lockdown as $\mathcal{R}_0 \approx 2.95$. After lockdown, this value declined to $\approx 0.68$, with a difference of $\approx 2.30$.

## SENSITIVITY ANALYSIS

In order to assess the relationship between our model parameters, we use the Latin hypercube sampling (LHS) technique, which is a scheme for simulating random parameter sets that adequately cover the parameter space (*Blower & Dowlatabadi, 1994*; *McGreal, 2020*; *Wang et al., 2013*). Uncertainty in model parameters can be identified through the Latin hypercube sampling technique, coupled with partial Rank correlation coefficients

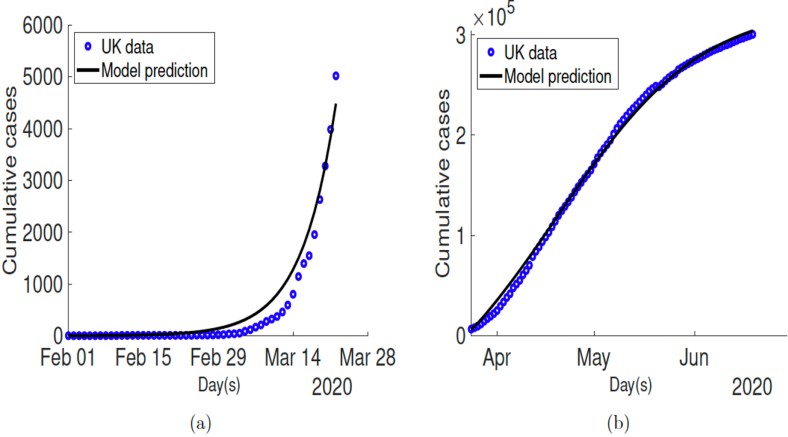

**Figure 2** **Time series plots showing the fitted model to COVID-19 related infectious cases for United Kingdom.** (A) Fitting before lockdown; (B) fitting after lockdown.

**Table 2** **Parameters values of the COVID-19 model (1) fitted to United Kingdom cumulative number of cases before lock down.** The data are obtained from Johns Hopkins website (*Dong, Du & Gardner, 2020*).

| Parameter | Description | Value | References |
|---|---|---|---|
| $\beta$ | Infection rate | 0.7566 | Fitted |
| $\eta_A$ | Asymptomatic infection rate modification parameter | 0.4352 | Assumed |
| $\eta_Q$ | Quarantined infection rate modification parameter | 0.135 | Assumed |
| $\eta_H$ | Hospitalized infection rate modification parameter | 0.3725 | Fitted |
| $q$ | Proportion developing asymptomatic infections | 0.6 | *Centers for Disease Prevention and Control (2020b)* |
| $\sigma$ | Disease progression rate | 0.60 | *Centers for Disease Prevention and Control (2020b)* |
| $\gamma_I$ | Recovery rates of symptomatic | 0.10 | *Agusto et al. (2022)* |
| $\gamma_A$ | Recovery rates of asymptomatic | 0.13978 | *Agusto et al. (2022)* |
| $\gamma_Q$ | Recovery rates of quarantined | 0.1 | *Agusto et al. (2022)* |
| $\gamma_H$ | Recovery rates of hospitalized | 0.0526 | *Centers for Disease Prevention and Control (2020b)* |
| $\omega_Q$ | Quarantine rate | 0.5679 | Fitted |
| $\omega_H$ | Hospitalization rate | 0.5180 | Fitted |
| $v_Q$ | Quarantine violation rate | 0.4638 | Fitted |
| $v_H$ | Hospital discharge rate | 0.1282 | Fitted |
| $\delta_I$ | Death rate of symptomatic | 0.0009 | *UK Health Security Agency (2022)* |
| $\delta_A$ | Death rate of asymptomatic | 0.00054 | *UK Health Security Agency (2022)* |
| $\delta_Q$ | Death rate of quarantined | 0.0009 | *UK Health Security Agency (2022)* |
| $\delta_H$ | Death rate of hospitalized | 0.0018 | *UK Health Security Agency (2022)* |

(PRCCs). We assume that each uncertain parameter is uniformly distributed within a specified range, which is within $\pm 30\%$ of the respective baseline parameter values, and performed a Latin hypercube sampling analysis by generating 1,000 random samples from the chosen parameter distributions. PRCCs were then calculated for each of the following parameters, $\gamma_A, \gamma_I, \gamma_Q, \gamma_H, \delta_A, \delta_I, \delta_Q, \delta_H, \sigma, q, \beta, \eta_A, \eta_Q, \eta_H, \omega_Q, \omega_H, v_Q, v_H$, and the

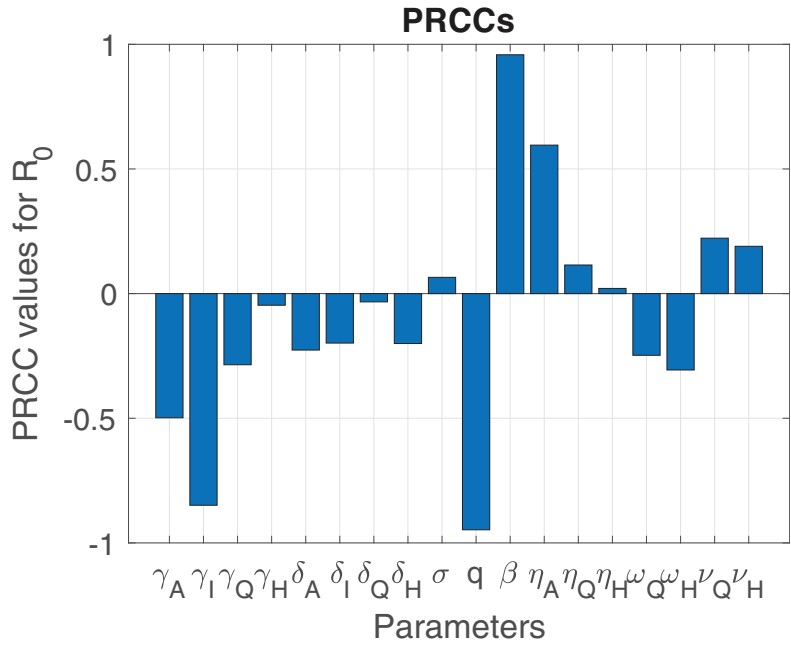

**Figure 3 Sensitivity analysis indicating PRCC results illustrating the dependence of $\mathcal{R}_I$ on the parameters of the model.**

outcome variable (the basic reproduction number, $\mathcal{R}_I$). The sign of the PRCCs indicates whether or not changes in the input parameter has a positive or negative effect on the corresponding output variable (*Wang, Liu & Heffernan, 2018*; *Wang, Liu & Liu, 2016*). The most influential parameters of the model are those that have PRCC values that satisfy $|PRCC| > 0.4$, where a negative sign indicates an inverse relationship. The correlation between the output variable and the input parameters is moderate if $0.2 < |PRCC| < 0.4$, and is weak otherwise (*Cariboni et al., 2007*).

Figure 3 indicate that the parameters $\beta, \eta_A, q, \gamma_I$, and $\gamma_A$ have the greatest impact on the outcome function (the reproduction number). On one hand, the parameters $\sigma, \eta_Q, \gamma_Q, \delta_A, \delta_Q, \delta_I, \omega_Q$ and $v_Q$ have a moderate impact on the reproduction number (the outcome function). The dominant parameters in increasing the outcome function ($\mathcal{R}_I$) are the transmission ($\beta$), and the infection modification parameter for the asymptomatic infectious ($\eta_A$). On the other hand, the dominant parameters in decreasing $\mathcal{R}_I$ are the proportion of exposed individuals developing infections ($q$), recovery rate of infectious ($\gamma_I$), mortality rate of infectious ($\delta_I$), the isolation rate of hospitalized and quarantined individuals ($\omega_H$ and $\omega_Q$, respectively).

## SENTIMENT ANALYSIS

In order to carry our the sentiment analysis, tweets from Twitter were downloaded from January 2, 2020 to May 29, 2020 for six countries, namely Australia, Brazil, Italy, South-Africa, United Kingdom, and United States of America, representing different geographical contexts. These countries and their population composition and political views are diverse, as are their sentiments about the virus. Different factors drive their

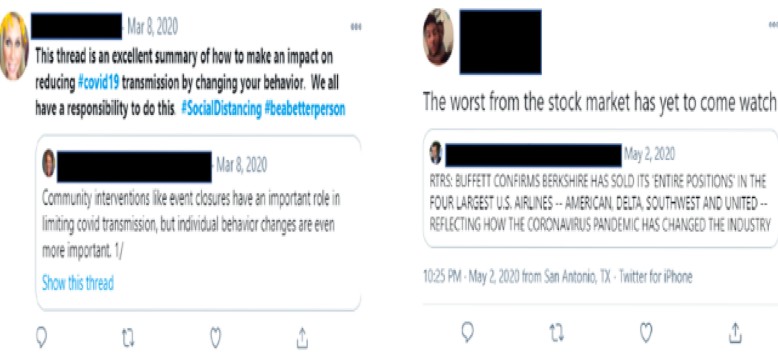

**Figure 4 Sample positive and negative tweets downloaded (A) sample positive tweet; (B) sample negative tweet.**

sentiments about the pandemic. For instance, in the United States, a common sentiment is that the virus is a hoax directed at the ruling party (*Oliver, 2020*; *Waldrop & Gallman, 2020*). Media reports about the outbreak are another factor driving the public sentiment but is being construed as fake news by some people. In Brazil, the president accused the press of spreading panic and paranoia (*BBC News Services, 2020*), and called the virus "a small flu" and urged the people to go to the streets and "face the disease like men" (*BBC News Services, 2020*; *Gray & Shapiro, 2020*; *Traumann, 2020*). Furthermore, the overall sentiment in Brazil is anti-science in nature where the president had once promoted the use of the antimalarial drug, hydroxycholoroquine, as coronavirus treatment drug despite lack of evidence that it was effective against the virus while rejecting social-distancing measures (*Fraser, 2020*; *Gray & Shapiro, 2020*).

In our work, we used the following procedure to generate sentiment scores for each country using COVID-19 tweets. Each tweet contains information including a unique tweet identification number (*i.e.*, tweetID) and text up to 240 characters as well as meta-information about the tweet such as user details, geographic origin, user-defined hashtags to categorize the tweet topic, language, and time of creation. In order to maintain consistency of extracting COVID-19 specific content with similar COVID-19 studies using twitter data, the tweetIDs were extracted from a public repository (*Chen, Lerman & Ferrara, 2020*). These tweetIDs contain validated tweets that include 76 hashtags related to COVID-19 including #COVID-19, #coronavirus, #Corona, #sars-cov-2, #Covid19, #SocialDistancing, #quarantinelife, #covididiot, *etc*. Figure 4 shows sample tweets downloaded over this time period. As evidenced, negative sentiment may be specific to the disease, expected behaviors, or the behaviors of other people. In part, this may reflect a form of polarization on disease responses.

The process of extracting tweets corresponding to the tweetIDs from the Twitter server is known as hydration, and was carried out by a verified Twitter developer with a valid application programming interface (API). The *Twarc* hydrator package in python was used to retrieve the tweets with a sleep time of one second between tweets to avoid the

**Table 3 Number of downloaded tweets from Twitter server by country from January 22 to May 29, 2020.**

| Country | Tweet count |
| --- | --- |
| Australia | 18,104 |
| Brazil | 27,684 |
| Italy | 55,816 |
| South Africa | 16,778 |
| UK | 99,080 |
| US | 661,567 |
| Total | 983,481 |

100,000 tweets/day extraction limit set by Twitter. A total of 125.2 million tweets were collected during January 22 to May 29, 2020 time period and stored in a google cloud platform (GCP) server with 8 GB RAM and 2 TB storage. However, less than 1% (983,481 tweets) of the total tweets downloaded was used, as most tweets do not have country indicator because very often users do not associate their account to a country and therefore remain anonymous in the geographic identities, see Table 3 for the country-by-country break down of downloaded tweets.

Post data collection, all tweets were translated to English using the *googletrans* package. Regular expressions were used for performing data cleaning operations on the tweet texts including removal of special symbols and filtering out URLs. Sentiment scores for all tweets were computed using the *textblob* package (*Ma, 2005*). The textblob python package used to compute the sentiment score for tweets in our study adopts a rule-based approach for sentiment quantification based on key indicator words present in them. A tweet is represented as a bag of words. The positive and negative sentiments of a sentences are based on the weighted average of annotated sentiments to each word in a large *corpus*. The sentiment/polarity score varies from $-1$ to $+1$, and if it is $<0$, we classify the tweet as a negative sentiment tweet, and $>0$ as a positive sentiment tweet. The tweet scores are then averaged for a country per day. Finally, for each of the above listed countries, the average positive sentiment per day was reported. Figure 5 shows the positive and negative sentiments for the respective countries.

To quantify the overall sentiment in each of the countries, we fitted straight lines ($y_p$ and $y_n$ for positive and negative sentiments) through these sentiments; and we took the difference of the lines $y_p$ and $y_n$ to determine if the overall sentiment from a country is positive or negative during the time period the tweets were collected. We see in Fig. 6 that Australia and the United Kingdom are the two most positive countries, their positive sentiment levels were really high. This is followed by Italy and South Africa which had moderately positive sentiment. Brazil and the United States of America have more negative sentiments overall, since they have the least positive sentiment. The European countries, although they were the first to experience a massive wave of the infection, remained relatively positive.
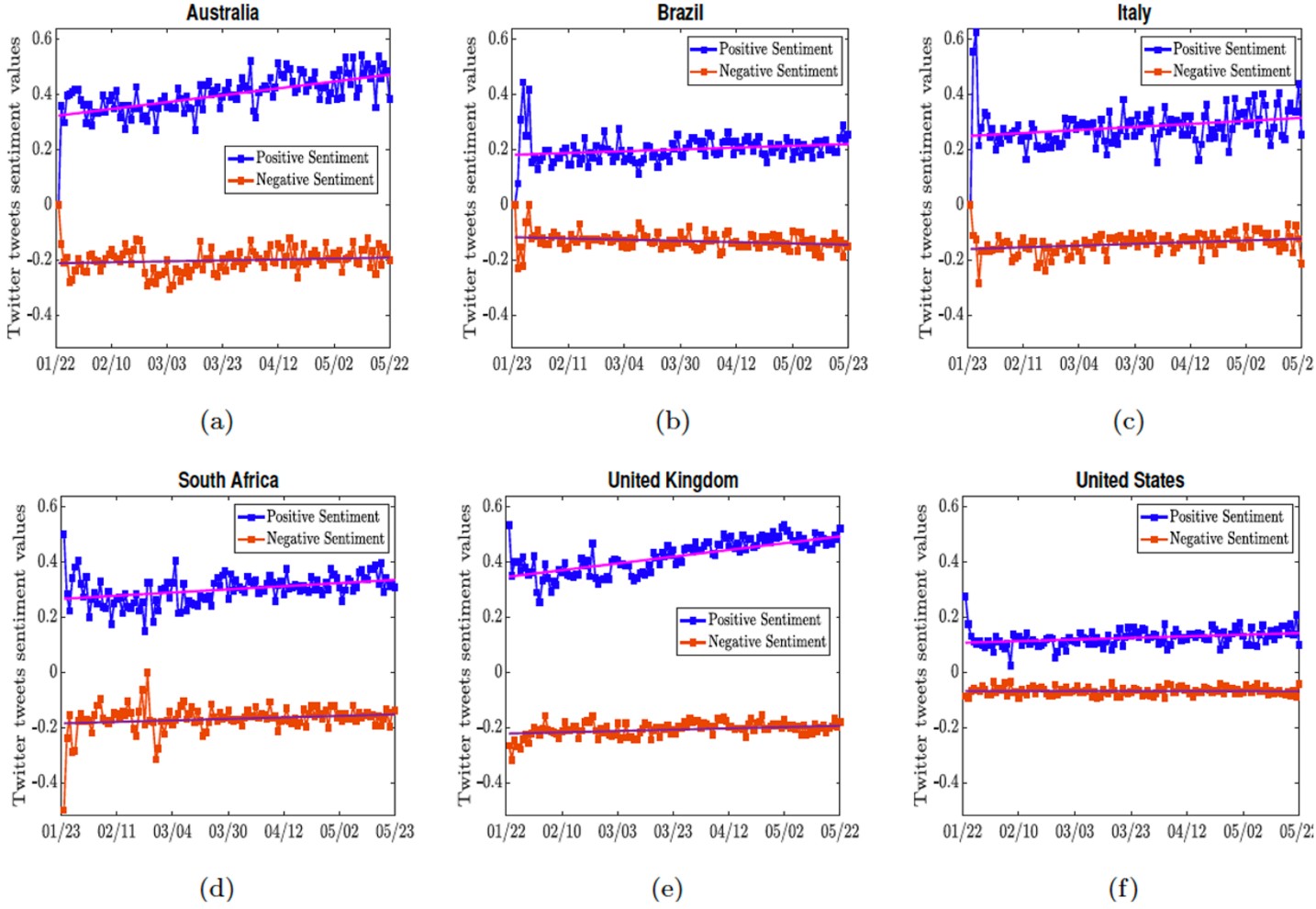

**Figure 5 Positive and negative sentiment of the COVID-19 tweets.** Two straight lines $y_p = a_p t + b_p$, and $y_n = a_n t + b_n$ are fitted through the positive and negative tweets. The straight lines for each country are given as (A) the lines $y_p = 0.0012461 \times t + 0.32225$, and $y_n = -0.00016767 \times t + 0.21212$ for Australia; (B) the lines $y_p = 0.00032631 \times t + 0.18091$, and $y_n = 0.00022551 \times t + 0.11779$ for Brazil; (C) the lines $y_p = 0.00054929 \times t + 0.24898$, and $y_n = -0.00030907 \times t + 0.16079$ for Italy; (D) the lines $y_p = 0.0005727 \times t + 0.26629, y_n = -0.00026964 \times t + 0.18524$ for South-Africa; (E) the lines $y_p = 0.0012266 \times t + 0.34568$, and $y_n = -0.0002375 \times t + 0.22246$ for United Kingdom (F) the lines $y_p = 0.00029309 \times t + 0.10708, y_n = 5.5321e - 06 \times t + 0.067976$ for United States.

## COVID-19 MODEL WITH SENTIMENT EFFECTS

In this section, we incorporated the public sentiments (positive and negative) into the COVID-19 model (1) using the fitted straight lines ($y_p$ and $y_n$ for positive and negative sentiments). First, we used the results obtained from the sensitivity analysis in "Sensitivity analysis" to determine the form of the sentiment driven functions.

Each of these six countries instituted lockdown measures as a way to control the spread of the virus. We expect that as public awareness increases due to increased media coverage of the infection and the lockdown mitigation efforts that public perception and sentiments will be positive, therefore leading to a decrease in disease transmission. We therefore expect the infection rate $\beta$ to be a decreasing function of public sentiment. However, we see from the sensitivity analysis that the infection rate $\beta$ would increase the reproduction

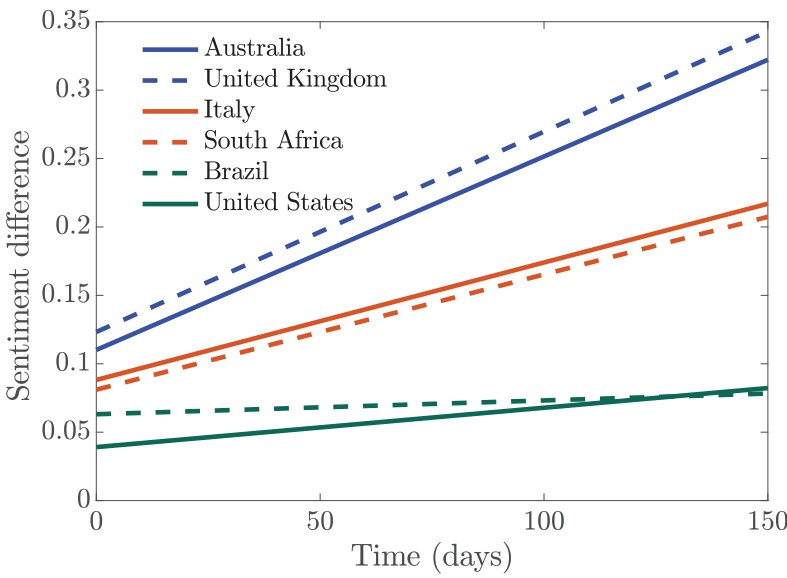

**Figure 6 Simulation of the difference between the straight line approximating positive and negative sentiments.**

number $\mathcal{R}_0$. Hence, we define a decreasing sentiment function for this parameter. We also define a decreasing sentiment-related function for $v_Q$ and $v_H$ since these parameters increase $\mathcal{R}_0$. However, the parameters $\omega_Q$ and $\omega_H$ are defined as increasing function of the perception-related functions. We have chosen these parameters because these parameters can be influenced by people's behavior, perceptions, and sentiments, unlike the recovery rates, $\gamma_A, \gamma_I, \gamma_Q, \gamma_H$, death rates, $\delta_A, \delta_I, \delta_Q, \delta_H$, disease progression rate, and the proportion asymptomatic, $q$. We discuss below how these functions are obtained for each of these parameters.

These countries instituted lockdown measures in March 2020 as a means to contain the virus. For instance Italy, Brazil, US, Australia, UK, and South Africa instituted lockdowns on March 9, March 17, March 19, March 23, March 23, and March 26, respectively. Therefore, we define functions that incorporate the values of these parameters before and after lockdown. Starting with the infection rate $\beta$, we define the sentiment-related function $\beta_M$ as

$$\beta_M = \beta_1 + (\beta_0 - \beta_1)e^{-mC_I(t)}, \tag{4}$$

where $\beta_0$, $\beta_1$ are the before and after lockdown infection rates. The variable $C_I$ is the cumulative number of symptomatic infectious individuals in the community; these are determined from the following equation.

$$C_I(t) = (1-q)\sigma \int_0^t E(\tau)d\tau.$$

Note that $C_I$ is not an epidemiological variable. Furthermore, $m$ induces the effect of public sentiment on reported cumulative number of infected cases in the community. If $m = 0$ or relatively small, the infection rates and $\beta$ are equal or close to the constant $\beta_0$. On the other

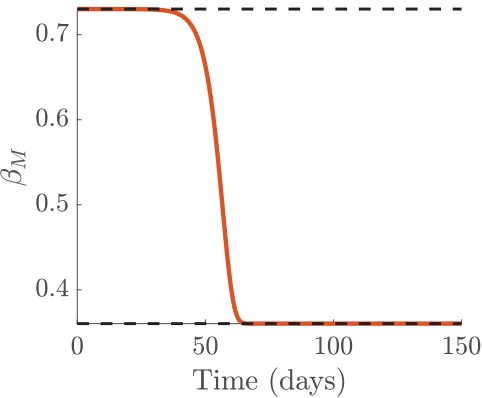

**Figure 7 Simulation of the functions showing media effect for the infection rate $\beta_M$, with media effect.**

hand, if $m > 0$, there is increase awareness about the disease in the community and the infection rate could decrease to $\beta_1(< \beta_0)$ as the number of accumulated infected cases increases as shown in Fig. 7.

Thus the public sentiment-related functions for quarantine ($\omega_{1M}$), hospitalization ($\omega_{2M}$), quarantine violation ($v_{1M}$), and early hospital discharge rate ($v_{2M}$) are represented by the following functions:

$$
\begin{aligned}
v_{QM} &= v_{Q1} + (v_{Q0} - v_{Q1})e^{-mC_I(t)} \\
v_{HM} &= v_{H1} + (v_{H0} - v_{H1})e^{-mC_I(t)} \\
\omega_{QM} &= \omega_{Q1} + (\omega_{Q0} - \omega_{Q1})e^{-mC_I(t)} \\
\omega_{HM} &= \omega_{H1} + (\omega_{H0} - \omega_{H1})e^{-mC_I(t)}.
\end{aligned}
\tag{5}
$$

Note that $v_{QM}, v_{HM}, \omega_{QM}, \omega_{HM} > 0$ for $C_I > 0$. We assume that $v_{Q1} < v_{Q0}, v_{H1} < v_{H0}$, and $\omega_{Q1} > \omega_{Q0}, \omega_{H1} > \omega_{H0}$. Furthermore, for arbitrarily small number of symptomatic infectious individuals $C_I$, the sentiment-related transition function $v_{QM}$ converges to $v_{Q0} > 0$ for small values of $C_I$ the maximum quarantine violation rate out of the quarantine class before the community lockdown. Also, as the cumulative number of infectious individuals $C_I$ grows, the quarantine violation function $v_{QM}$ converges to $v_{Q1}$, that is,

$$
\lim_{C_I \to \infty} v_{QM} = v_{Q1} > 0,
$$

the minimum quarantine violation rate out of the quarantine class as public perceptions and sentiments effects of the infection manifest in the community.

Similarly, the sentiment-related early hospital discharge rate, $v_{HM}$, from the hospitalized class, converges to $v_{H0} > 0$, the maximum early discharge rate for small cumulative number of infectious individuals $C_I$ before the onset of public perceptions and sentiments about the disease, and

$$
\lim_{C_I \to \infty} v_{HM} = v_{H1} > 0,
$$

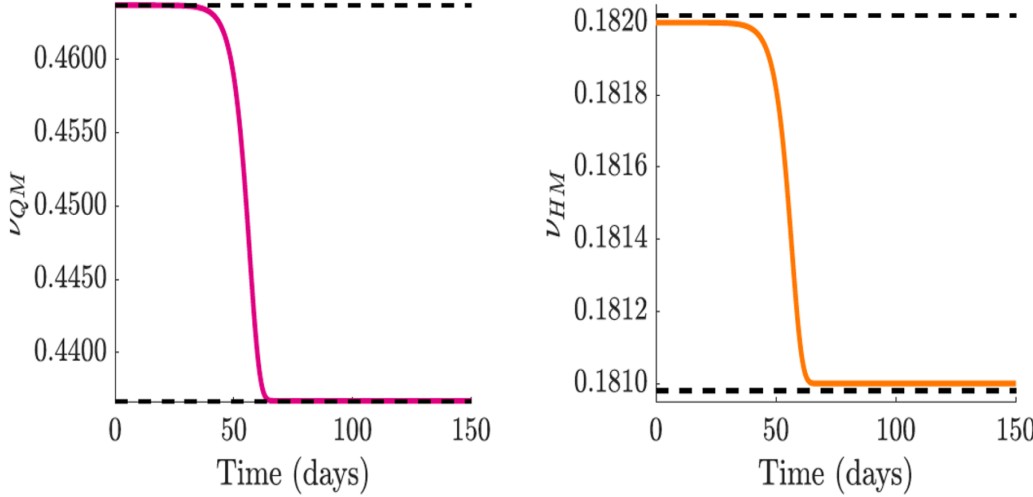

**Figure 8 Simulation of the quarantine violation, $v_{QM}$, and early hospital discharge, $v_{HM}$ functions which incorporate public sentiments.** Since $v_Q$ and $v_H$ increase $\mathcal{R}_0$, we used decreasing functions with sentiment that will reduce $\mathcal{R}_0$. (A and B) Quarantine violation and early hospital discharge rate $v_{QM}$ and $v_{HM}$, with the effect of public sentiments.

the minimum early discharge rate for large cumulative number of infectious individuals $C_I$ as public perceptions and sentiments effects manifest. The dynamic behavior of functions $v_{QM}$ and $v_{HM}$ are shown in Figs. 8A and 8B.

Consequently, for an arbitrarily small cumulative number of infectious individuals $C_I$, the public sentiment-related quarantine and hospitalized functions $\omega_{QM}$ and $\omega_{HM}$ converge to $\omega_{Q0} > 0$, and $\omega_{H0} > 0$, the minimum quarantine rates before the onset of public awareness. Also, as the cumulative number of infectious individuals $C_I$ gets larger, $\omega_{QM}$ converges to $\omega_{Q1}$ and $\omega_{HM}$ converges to $\omega_{H1}$. That is,

$$\lim_{C_I \to \infty} \omega_{QM} = \omega_{Q1} > 0, \quad \text{and} \quad \lim_{C_I \to \infty} \omega_{HM} = \omega_{H1} > 0$$

the maximum number of individuals that are self-isolated or hospitalized, respectively, as a result of media coverage. See Figs. 9A and 9B for the dynamic behavior of functions $\omega_{QM}$ and $\omega_{HM}$.

The sentiment parameter $m$ is expressed as $m = \frac{1}{\varepsilon}(y_p + y_n)$, where $y_p$ is positive sentiments, $y_n$ is negative sentiments, and $\varepsilon$ is a scaling factor that scales the sentiments per 100,000 of the population density. As described above, we fitted two straight lines through the positive and negative sentiments for each of the countries (see Fig. 5) to obtain the sentiment variable $y_p$ and $y_n$ given as

$$\begin{aligned} y_p &= a_p t + b_p \\ y_n &= a_n t + b_n, \end{aligned} \tag{6}$$

where $a_p$ and $a_n$ are the slope of the straight lines and $b_p$ and $b_n$ are the intercept.

Now, incorporating the sentiment-related functions (4), (5), and twitter sentiments (6) into the COVID-19 model (1), we have the following system of differential equations

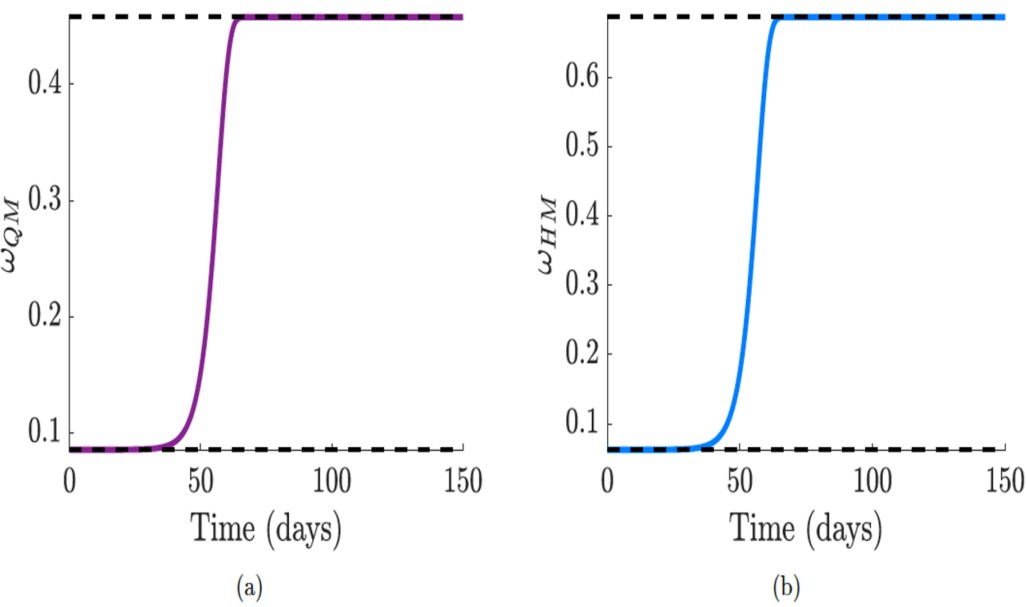

**Figure 9 Simulation of the public sentiment-related quarantine and hospitalization functions.** Since $\omega_Q$ and $\omega_H$ decreases $\mathcal{R}_0$, we used increasing functions incorporating sentiments that will reduce $\mathcal{R}_0$. (A and B) Quaratine, $\omega_{QM}$, and hospitalization, $\omega_{HM}$ functions with public sentiments.

$$\frac{dS}{dt} = \frac{-\beta_M[I(t) + \eta_A A(t) + \eta_Q Q(t) + \eta_H H(t)]S(t)}{N(t)}$$

$$\frac{dE}{dt} = \frac{\beta_M[I(t) + \eta_A A(t) + \eta_Q Q(t) + \eta_H H(t)]S(t)}{N(t)} - \sigma E(t)$$

$$\frac{dA}{dt} = r\sigma E(t) - \gamma_A A(t)$$

$$\frac{dI}{dt} = (1-r)\sigma E(t) + v_{QM}(t)Q(t) + v_{HM}(t)H(t) - \omega_{QM}(t)I(t) - \omega_{HM}(t)I(t) - \delta_I I(t) \qquad (7)$$

$$\frac{dQ}{dt} = \omega_{QM}(t)I(t) - \gamma_Q Q(t) - v_{QM}(t)Q(t) - \gamma_Q Q(t)$$

$$\frac{dH}{dt} = \omega_{HM}(t)I(t) - \gamma_H H(t) - v_{HM}(t)H(t) - \gamma_H H(t)$$

$$\frac{dR}{dt} = (\gamma_I + \delta_I)I(t) + (\gamma_A + \delta_A)A(t) + (\gamma_Q + \delta_Q)Q(t) + (\gamma_H + \delta_H)H(t)$$

$$y_p = a_p t + b_p$$

$$y_n = a_n t + b_n.$$

The reproduction number related to model (7) with Twitter sentiment is given as

$$\mathcal{R}_{0T} = \frac{(1-q)\beta_0(K_3\eta_H\omega_{H_0} + K_4\eta_Q\omega_{Q0} + K_3 K_4)}{(K_2 K_3 K_4 - K_3 v_{H0}\omega_{H0} - K_4 v_{Q0}\omega_{Q0})} + \frac{q\beta_0\eta_A}{k_1}, \qquad (8)$$

where, $K_2 = l_2 + \omega_{H_0} + \omega_{Q_0}$, $K_3 = l_3 + v_{Q_0}$, $K_4 = l_4 + v_{H_0}$, $l_1 = \gamma_A + \delta_A$, $l_2 = \gamma_I + \delta_I$, $l_3 = \gamma_Q + \delta_Q$, $l_4 = \gamma_H + \delta_H$.

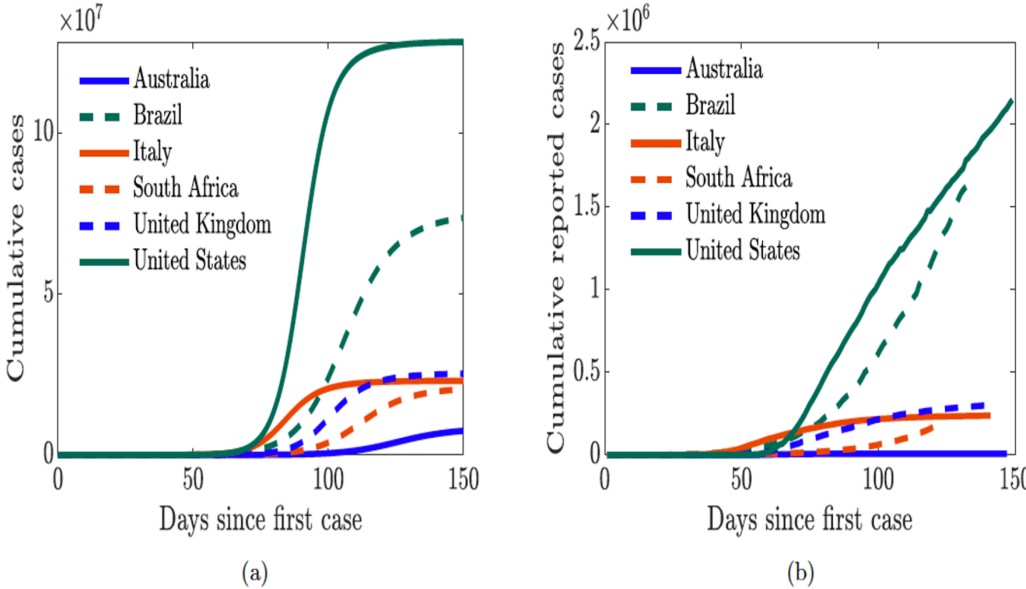

**Figure 10 Simulation of the cumulative cases from COVID-19 model (7) with public sentiment and the cumulative reported cases from January–May 2020.** (A) Simulated cumulative cases obtained from the COVID-19 model (7) with public sentiments; (B) cumulative number of reported cases.

The reproduction number, $\mathcal{R}_{0T}$, is the average number of secondary infectious produced when a single infected individual is introduced into a completely susceptible population.

Next, we simulated the sentiment-related model (7) using the estimated parameters for each country and plotted in Fig. 10A the cumulative new cases for each of the countries and compare the results to the trajectory of the actual cumulative reported cases in Fig. 10B. We see that the sentiment-related model (7) accurately captures the trajectory of the actual cumulative reported cases; therefore indicating that incorporating public sentiment into an epidemic model is able to capture the trend in the trajectory of the infection in the population. Although the model-simulated cumulative number of cases saturates much earlier than the actual cumulative number of cases; at this point, we are not sure why. Nevertheless, we are able to realize our goal of understanding the role of public sentiment in disease spread since we are not using the sentiment-related model (7) to make prediction about the number of cases.

## RESULTS

We begin by analyzing the COVID-19 transmission model with quarantine and hospitalization coupled with public sentiment (described in "COVID-19 model with sentiment effects"). Then we analyze the effect of public sentiment and human behavior on the spread and prevalence of COVID-19 in the community.

### Impact of public sentiments on disease transmission

In this section, we explore the impact of public sentiments (positive or negative) on disease transmission in the population. Using sentiment-related functions parameterized with

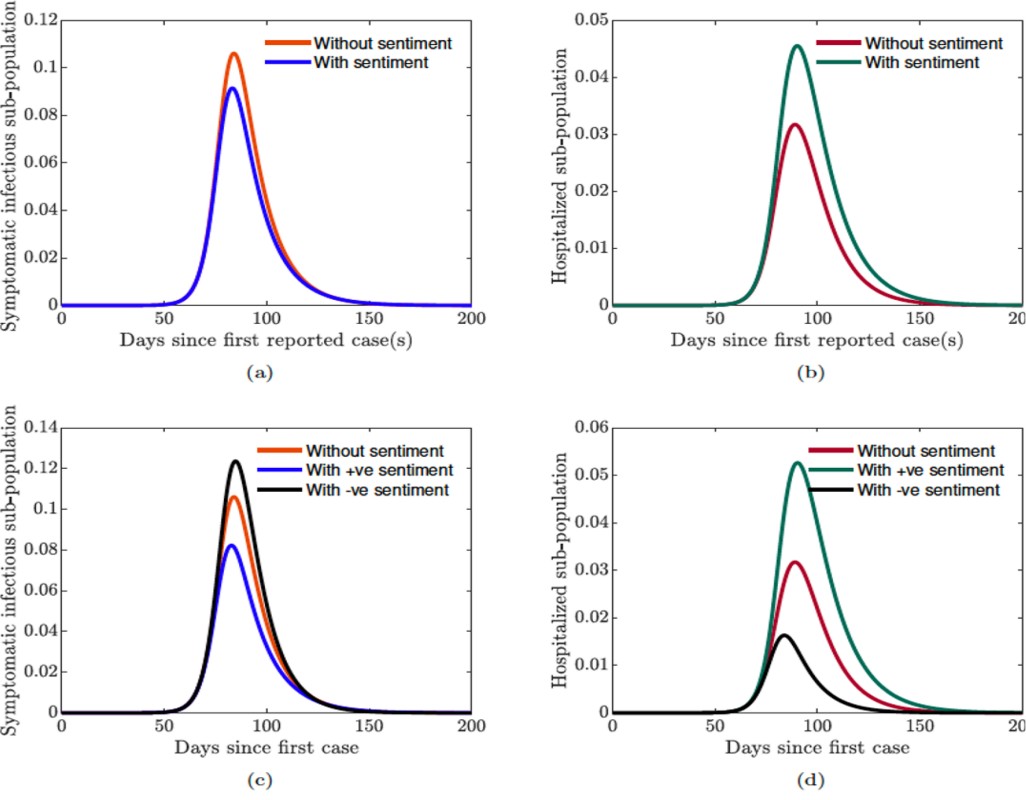

**Figure 11 Simulation of the sentiment-related COVID-19 model (7).** (A and C) Symptomatic infectious individuals without/with sentiments; (B and D) hospitalized sub-population with sentiments (positive and negative) against no sentiments.

data from United Kingdom, Fig. 11 shows the impact of incorporating public sentiments in the model against a model without sentiments. We see from Fig. 11A that the number of symptomatic infectious individuals in the population is lower when we incorporate positive sentiment into the COVID-19 model. On the other hand, we see a higher number of symptomatic infectious individuals in the population when public sentiments are not included in the model. However, in Fig. 11B, we see that the number of hospitalized individuals in the population is higher with positive sentiment. And a lower number of hospitalized individuals in the population when public sentiments are not included in the model. Furthermore, we see in Fig. 11C that negative public sentiments will yield even more symptomatic infectious individuals in the population, but fewer hospitalized individuals in the population (see Fig. 11D). The result involving the hospitalized, show a counter intuitive result, as one would expect to see more hospitalization with cases. However, with negative sentiment comes mistrust in establishments. Thus, it makes sense if we are seeing fewer infectious individuals seeking hospitalized treatment. During the outbreak in 2020 many individuals in the United States relied on chloroquine and hydroxychloroquine, two drug treatment for malaria as treatment for COVID-19 and would only go to the hospital when they are critically ill (*Joseph et al., 2005*; *Mahmood, 2020*; *US Food and Drug Administration, 2020e*).

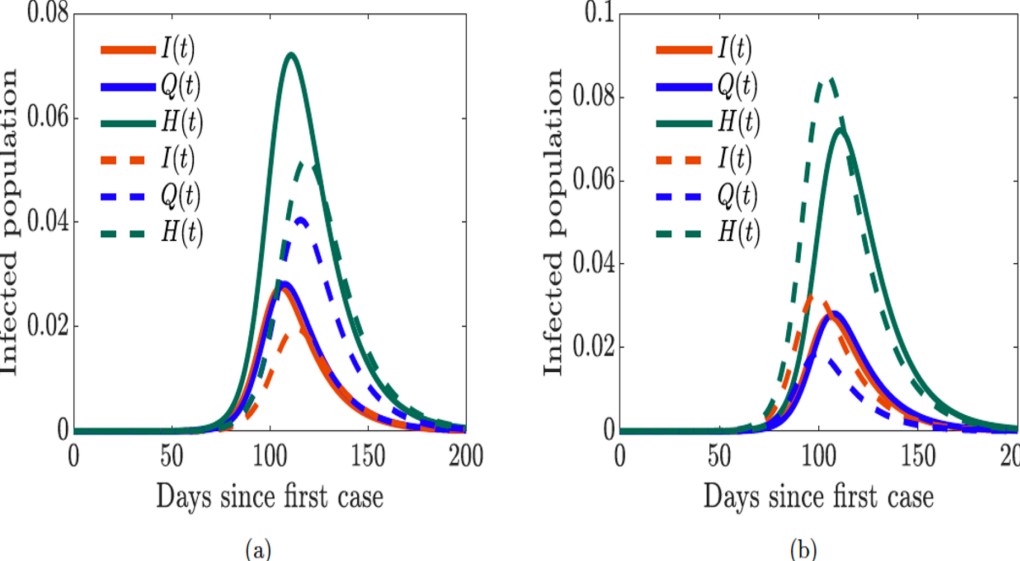

**Figure 12 Simulation of the sentiment-related COVID-19 model (7) for the proportions of symptomatically infected (I), quarantined (Q), and hospitalized (H) individuals.** Solid lines correspond to base values of the model parameters from Table 2. (A) Dashed lines correspond to double quarantine ($\omega_Q$) and hospitalization ($\omega_H$) rates (B) dashed lines correspond to double quarantine violation ($v_Q$) and hospital discharge ($v_H$) rates.

Thus, the results shown in Fig. 11 suggests that it is important to incorporate public sentiments into epidemic models. Having a clear understanding of the public perception of the risk of the infection and their sentiments regarding a disease outbreak and its transmission is vital for control and mitigation efforts.

## Impact of human behavior on quarantine and hospitalization

Next, we explore the impact of quarantine and hospitalization on the number of hospitalized individuals in the population while using the sentiment-related functions parameterized with data from United Kingdom. First, we double the quarantine and hospitalization rates ($\omega_Q$ and $\omega_H$). We notice in Fig. 12A that the epidemic curve for the hospitalized individuals increases and the peak of the curve shifts from left to right (as do the time the infection peaks); similarly, the symptomatic infectious individuals shrink ("flatten the curve") while the quarantined population increases and their curves shifts to the right since the rates have been increased. However, when we double the quarantine violation and early hospital discharge rates ($v_Q$ and $v_H$), we see in Fig. 12B the curve of the hospitalized individuals shifts from right to left and the number of hospitalized individuals increases. We see similar shifts in dynamics of the symptomatic infectious and quarantined individuals; however, there are fewer symptomatic infectious and quarantined individuals in the respective classes. It is thus vital to ensure public compliance and adherence with quarantine rules and to promote positive sentiment among the populace, as this will go a long way in flattening the epidemic curve.

## DISCUSSIONS AND CONCLUSIONS

### Discussions

In this study, we developed a novel compartmental mathematical model to study the ongoing COVID-19 pandemic. The model uniquely incorporated human behavior and early discharge from hospital. The model is further coupled with public sentiments about the disease, thereby capturing the effect of disinformation. In particular, the model includes violation of quarantine rules and their positive and negative sentiments regarding the disease. The model also includes discharge of the infected due to overwhelmed hospital facilities. For instance, at the onset of pandemic in England, seniors in hospitals were moved back to care homes (*Pawelek, Oeldolf-Hirsch & Rong, 2014*; *Servick, 2020*). Similarly, at the height of the outbreak in Michigan and New York, hospitals were discharging the non-critically ill either to nursing homes or simply letting them go home because the hospital facilities were overwhelmed (*Boucher, 2020*; *NBC 25 News, 2020*; *Schnirring, 2020*), prompting legislation in Michigan to protect the seniors and vulnerable members of the community and prevent nursing homes from admitting patients with COVID-19 (*Newport, 2020*). In other places like Arizona, some nursing homes took in COVID-19 patients with mild symptoms (*Crenshaw, 2020*).

Public awareness and information is one of the factors driving public perception of risk and sentiment about the disease (*Harvard Mental Health Lettert, 2020*; *Ong et al., 2020*). At the onset of the pandemic, many people believed (unfortunately, many still believe) that the virus was a hoax, along with wide range of other conspiracy theories about the disease (*Andersen, 2020*; *Cahn, 2020*; *Galbraith, 2020*; *Imhoff & Lamberty, 2020*; *Miller, 2020*; *Specia, 2020*). Many of these misconceptions and disinformation about the disease were spread on social media platforms like Twitter, Facebook, *etc.*, which in turn drives public views, opinions and sentiments about the disease (*Alamoodi et al., 2021*; *Jarynowski, Wojta-Kempa & Belik, 2020*). For instance, *Jarynowski, Wojta-Kempa & Belik (2020)* using Twitter was able to capture in Poland the structural division of the Polish political sphere, identifying the mainstream opposition and protestant groups, and the possible orgin of disinformation in the country. In Brazil, the prevalence of misinformation surrounding the pandemic is deeply concerning and many people blame the messaging from the President Bolsonaro (*Gray & Shapiro, 2020*).

To measure public sentiments across six countries across different geographical regions, we downloaded tweets from the Twitter platform from January to May 2020. We then carried out sentiment analysis that enabled us to separate the public sentiment into either positive or negative sentiment. While our data set is a multilingual data set across multiple countries, the filter keywords (*i.e.*, hashtags) are mostly in English. Even though 76 hashtags have been considered to extract tweets, there exists a possibility of excluding tweets that pertain to COVID-19 but do not contain any of these hashtags. Even though we have carried out basic data cleaning and processing tasks, we may have overlooked the small proportion of tweets that contain regional phrases expressing irony that may not all have been discovered by the sentiment analysis software program that explores the English translated text. However, a visual inspection of the texts and the sentiment scores for each

tweet across the different countries showed that the sentiment scores were representative of the tweet sentiment in most cases, thus ruling out systematic biases in inferences made in this study. The collection, aggregation, and analysis of more than 100 million COVID-19 related tweets generated during the time period ensures representation of the general public sentiment across all sub-populations of each country and not just one region or demographic segment.

Misinformation, disinformation, and conspiracy theories can be really problematic with tremendous impact on public health efforts to contain the disease in the community. However, the use of Twitter tweets to measure public sentiment may be limiting and not present the full picture of public sentiment since public information campaigns might have less impact on society than expected due to filter "bubbles" observed on Twitter (*Jarynowski, Wojta-Kempa & Belik, 2020*). Hence, it will be beneficial to diversify the sources of public awareness and information in other to reach many people as possible (*Jarynowski, Wojta-Kempa & Belik, 2020*) and possibly reduce the spread of disinformation.

After obtaining the positive and negative sentiments, we fitted straight lines through the sentiments in order to determine the magnitude of the sentiments in each of the countries, see Fig. 6. We see that United States and Brazil had the least positive sentiment. The level of public sentiments in the United States may be due in part to how polarized the country was in the last 4 years particularly in the months leading to the 2020 presidential elections. United Kingdom and Australia had very positive sentiment overall; in the early days of the pandemic in the UK, the entire country including the royal family applauded the selfless efforts of the health workers and other frontline workers (*BBC News, 2020a*), sharing clips on social media under the #ClapForCarers hashtag (*Aljazeera, 2020*; *Saini, 2020*).

On March 11, 2020 the World Health Organization (WHO) declared the novel coronavirus a global pandemic (*WHO, 2020c*) and shortly thereafter many countries imposed travel bans from many hotspots regions, and instituted lock-down measures in a bid to curtail and contain the spread of the virus. To incorporate public sentiment into our COVID-19 model (1), we segmented the time period into before and after lockdown. We used results obtained from the sensitivity analysis (see Fig. 3) to informed the nature of the different parameters (see Figs. 7–9) that can be influence by public sentiment. These parameters were then defined as increasing and decreasing functions of pubic sentiment which were incorporated into COVID-19 model (1). These parameters consist of before and after lockdown related parameters which we estimated using data are obtained from Johns Hopkins website (*Dong, Du & Gardner, 2020*) and some parameter values from literature (see Table 2).

The reproduction numbers for before and after lockdown ($\mathcal{R}'_l$ and $\mathcal{R}'_\infty$) for the respective countries are shown in Table A2. All the countries had reproduction number above one before lockdowns were put in place, and the values were below one after lockdown except for South Africa with estimated value of $\mathcal{R}'_\infty = \infty . \triangledown \in$. This value aligns with the estimated value by the South African National Institute for Communicable Diseases (*South African National Institute for Communicable Diseases, 2020*).
The reproduction number estimated for South Africa by the National Institute for

Communicable Diseases at the onset of the outbreak was between 1.7 and 2.5; these numbers reduced substantially but was still above one following measures such as flight restrictions into the country, school closures, and national level 5 lockdown in mid-March 2020. Some provinces like Western Cape Province had estimated reproduction number of 1.5–1.7 by mid-late April 2020, while other provinces like Gauteng, KwaZulu Natal, and Eastern Cape Province had estimated reproduction number of 1.0–1.5 by mid-late April 2020 indicating an ongoing transmission or steady disease progression (*South African National Institute for Communicable Diseases, 2020*).

Our results showed that preventing the spread of disinformation and negative sentiment about the disease in the community is important (see Fig. 11). Thus, it is essential to prevent disinformation, and to promote positive sentiment in the community. It is equally vital to ensure public compliance and adherence with quarantine rules and all mitigation efforts (see Fig. 12). Doing so will go a long way in flattening the epidemic curve, and will lead to the kind of success story observed in New Zealand (*Shepherd, 2020*; *Wikipedia, 2020*).

Our study demonstrated that the countries with positive sentiment, and quarantine compliance have been more successful at curtailing the spread of the disease. In addition, we have been able to demonstrate the impact on disease burden of early discharge of symptomatic infectious individuals from hospital to make room for incoming sever COVID-19 patients. Overall, our model is able to demonstrate the role of people's behavior and public sentiment on disease transmission. Although, the trajectory of model simulation in Fig. 10A is able to capture the trend of the actual trajectory of the cumulative number of cases in Fig. 10B, our simulation results saturate much earlier. A number of factors may be responsible for this, for instance, non ascertainment of all infected cases. According to CDC (*Centers for Disease Prevention and Control, 2020b*), asymptomatic individuals can account for between 15% to 70% of cases which in reality are not tracked nor documented. Note that our model in Fig. 1 incorporate the asymptomatic individuals this may be the reason for the difference between the simulated outcome on the case data.

Since we started this study, the number of cases in these countries has exploded, with some experiencing multiple waves of infections (*WHO, 2020b*) put in another lock-down (France, Germany, Italy, and the United Kingdom (*BBC News, 2020b*, *2020c*; *Levy et al., 2017*; *Meloni & Hutchinson, 2020*; *Savage, 2020*)). Although we did not evaluate the sentiment after the lock-down was lifted, we observed a wave of protests against other mitigation efforts like the use of face-mask and vaccines in many of these countries such as US, UK, Australia, Italy, and Canada (*Drury, 2020*; *McGee, Reynolds & Cullen, 2020*; *Reuters, 2020*; *Rinke & Kar-Gupta, 2020*). We believe these protests are driven by negative sentiments in the society against the use of face-masks which subsequently increases the number of infection as we observed in Fig. 11.

## CONCLUSION

To conclude, this study develops a novel model for COVID-19 that uniquely incorporates human behavior driven by their perception of risk and sentiments about the disease.

The goal of this study was not to make explicit epidemiological predictions about the disease; rather we hope to provide insight into effects of human behavior on non-pharmaceutical intervention strategies (such as self-isolation and quarantine) aimed at containing the disease and public sentiments about the disease. The key findings from this study are summarized below.

The simulations of the COVID-19 model (7) with human behavior and public sentiment about the disease show that:

i) Incorporating public sentiment into an epidemic model is able to project the trajectory of the disease incidence in the community.

ii) Positive sentiments among individuals in the population reduces the number of infected and disease burden in the community.

iii) Negative sentiments among individuals in the community amplify the disease burden in the community.

iv) Increasing quarantine, and hospitalization rates decreases the disease burden and reduces epidemic peak.

v) Increased quarantine violation rate and early discharged of those still infectious due to overwhelmed hospital resources increases disease burden leading to early epidemic peak.

This study has shown that incorporating human behavior and public sentiment into epidemic models is pertinent in order to accurately capture the dynamics and burden of the disease in the community. We have seen the role quarantine violation plays in disease spread; in a future study, we will incorporate other kinds of mitigation efforts such as vaccination and public reactions about them. Aside for incorporating mitigation efforts, in our future model we will consider the hospital capacity in terms of the number of bed. At the height of the outbreak a number of hospitals both in urban and rural areas exceeded their capacity to accommodate infected individuals.

## A PARAMETER ESTIMATION FOR THE SELECTED COUNTRIES

Initial values for our simulations are given in Table A1, it include the population of the countries $N(0)$ and the exact cumulative value $C(0)$ from the data. The initial values of $E_0$, $A_0$, $I_0$, $H(0)$, and $R(0)$ to ensure the fit of the trajectory of each country. The initial values are summarized below:

**Table A1 Values of the initial conditions used for the fitting and the objective function $J$ given in (3).**

| Countries | N(0) | E(0) | A(0) | I(0) | Q(0) | H(0) | R(0) | C(0) | $J_0$ | $J_1$ |
|---|---|---|---|---|---|---|---|---|---|---|
| Australia | 25,499,884 | 0 | 3 | 0 | 0 | 0 | 0 | 3 | 0.52 | 0.029 |
| Brazil | 212,559,417 | 100 | 100 | 1 | 0 | 0 | 110 | 1 | 0.10 | 0.06 |
| Italy | 60,461,826 | 1 | 10 | 2 | 1 | 0 | 0 | 2 | 0.13 | 0.04 |
| S. Africa | 59,355,826 | 1 | 1 | 218 | 1 | 100 | 0 | 927 | 0.09 | 0.92 |
| UK | 67,988,148 | 9,860 | 48,000 | 967 | 9,860 | 9,560 | 9,760 | 6,654 | 0.17 | 0.02 |
| US | 331,002,651 | 5 | 10 | 1 | 0 | 0 | 5 | 1 | 0.16 | 0.05 |

**Table A2 Fitted parameters values before and after lock down for Australia, Brazil, Italy, South Africa, United Kingdom, and United States.** The constant parameters used for the model fitting are $q = 0.6$, $\sigma = 0.6$, $\eta_Q = 0.135$, $\eta_A = 0.4352$, $\gamma_A = 0.13978$, $\gamma_I = 1/10$, $\gamma_Q = 1/10$, $\gamma_H = 1/19$. The parameters $\beta_0 = 0.5713^*, 0.4656^{**}$ was used for Brazil, and South Africa in Fig. 10.

| Parameters | Australia | Brazil | Italy | S. Africa | UK | US |
|---|---|---|---|---|---|---|
| $\beta_0$ | 0.5682 | 0.4713* | 0.7027 | 0.7360** | 0.7738 | 0.7430 |
| $\eta_{H0}$ | 0.3725 | 0.5869 | 0.5948 | 0.3660 | 0.2131 | 0.4785 |
| $v_{Q0}$ | 0.3522 | 0.4363 | 0.5831 | 0.2771 | 0.4638 | 0.4700 |
| $v_{H0}$ | 0.3101 | 0.2649 | 0.3455 | 0.1781 | 0.1282 | 0.2799 |
| $\omega_{Q0}$ | 0.4467 | 0.4751 | 0.3548 | 0.0568 | 0.5679 | 0.2520 |
| $\omega_{H0}$ | 0.4980 | 0.2380 | 0.5237 | 0.0618 | 0.5180 | 0.1301 |
| $\delta_A$ | 0.00039 | 0.00056 | 0.00081 | 0.00056 | 0.00054 | 0.00045 |
| $\delta_I$ | 0.0006 | 0.00093 | 0.00135 | 0.00093 | 0.0009 | 0.00075 |
| $\delta_Q$ | 0.00065 | 0.00093 | 0.00135 | 0.00093 | 0.0009 | 0.00075 |
| $\delta_H$ | 0.0013 | 0.00185 | 0.0027 | 0.00185 | 0.0018 | 0.0015 |
| $\beta_1$ | 0.1916 | 0.6534 | 0.3017 | 0.4061 | 0.3490 | 0.2319 |
| $\eta_{H1}$ | 0.1206; | 0.2348 | 0.2461 | 0.5475 | 0.2988 | 0.8418 |
| $v_{Q1}$ | 0.1345 | 0.2347 | 0.1258 | 0.1100 | 0.2178 | 0.1305 |
| $v_{H1}$ | 0.3011 | 0.2202 | 0.3155 | 0.1576 | 0.1252 | 0.1110 |
| $\omega_{Q1}$ | 0.6941 | 0.4820 | 0.6792 | 0.2411 | 0.5776 | 0.5350 |
| $\omega_{H1}$ | 0.5242 | 0.5979 | 0.5372 | 0.1098 | 0.5281 | 0.1610 |
| $\mathcal{R}_0^0$ | 2.4211 | 2.1292 | 3.5972 | 3.7801 | 2.946 | 3.6126 |
| $\mathcal{R}_1^0$ | 0.5930 | 0.8806 | 0.6450 | 1.5173 | 0.6843 | 0.8477 |

The initial susceptible is given as $S(0) = N(0) - E(0) - A(0) - I(0) - Q(0) - H(0) - R(0) - C(0)$.

### Funding

This research was supported by the National Science Foundation under the grant number DMS 2028297. Enahoro A. Iboi had additional support from the National Science Foundation with award number #176194. The funders had no role in study design, data collection and analysis, decision to publish, or preparation of the manuscript.

### Grant Disclosures

The following grant information was disclosed by the authors:
National Science Foundation: DMS 2028297, #176194.

### Competing Interests

The authors declare that they have no competing interests.

## Author Contributions

- Folashade B. Agusto conceived and designed the experiments, performed the experiments, analyzed the data, prepared figures and/or tables, authored or reviewed drafts of the article, and approved the final draft.
- Eric Numfor performed the experiments, analyzed the data, prepared figures and/or tables, and approved the final draft.
- Karthik Srinivasan performed the experiments, analyzed the data, prepared figures and/or tables, authored or reviewed drafts of the article, and approved the final draft.
- Enahoro A. Iboi performed the experiments, analyzed the data, prepared figures and/or tables, and approved the final draft.
- Alexander Fulk curated the data and approved the final draft.
- Jarron M. Saint Onge conceived and designed the experiments, authored or reviewed drafts of the article, and approved the final draft.
- A. Townsend Peterson conceived and designed the experiments, authored or reviewed drafts of the article, and approved the final draft.

## Data Availability

The data is available at GitHub: https://github.com/FBAGUSTO/COVID-19_Sentiment_Effects.git.

The COVID-19 case data are available at DOI 10.1016/S1473-3099(20)30120-1.

The Twitter data are available at Chen E., Lerman K., Ferrara E.

Tracking Social Media Discourse About the COVID-19 Pandemic: Development of a Public Coronavirus Twitter Data Set. JMIR Public Health Surveillance 2020;6(2):e19273 DOI 10.2196/19273.

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
