# Peer review of "Impact of public sentiments on the transmission of COVID-19 across a geographical gradient"

_PeerJ, doi:10.7717/peerj.14736_

## Round 0.1 · original submission · Major Revisions

Both reviewers expressed substantial concerns to drastically revise the manuscript. Reviewing published studies in the Introduction would be an essential step to revise your manuscript. Reviewer 1 suggested removing one of your two models so please carefully consider how you address this comment. Reviewer 2 pointed out the potential statistical flaw in using the cumulative number of cases, which I must contend.

Reviewer 1 ·

Basic reporting

please see attached report.

Experimental design

please see attached report.

Validity of the findings

please see attached report.

Additional comments

please see attached report.

Annotated reviews are not available for download in order to protect the identity of reviewers who chose to remain anonymous.

Reviewer 2 ·

Basic reporting

no comment

Experimental design

no comment.

Validity of the findings

1. Model fitting is conducted using the cumulative number of cases in each country by least square approach. Are the estimated parameters reasonable compared with alternatively available estimates or observed data? It is very interesting to me that those who are asymptomatic result to death by 0.001. Moreover, the same value is estimated for those quarantined and hospitalized which further confuses me. Can the death rate of asymptomatic and hospitalized be same? I am not convinced that minimizing the least squares for 12 parameters would lead to a valid estimate, and validation of the derived estimates are necessary.

2. On the supplementary material, the R is estimated before and after the lockdown. The R after lockdown is below 1 for only Australia and US which means the epidemic continuously expanded despite lockdown for other countries. Did you validate this with observed incidence data? Further, it would be helpful if you can define if it is the basic reproduction number or effective reproduction number.

3. Is there any reason to observe a linear trend for the sentiments? Is there any difference in the trend before and after lockdown? Does the trend in sentiment correlate with other observable measures (e.g. mobility)?

Additional comments

The authors explore the impact of public sentiments on the transmission of COVID-19 in 6 different countries by extracting tweets from an online repository and scoring the sentiments after translating to English. Before mass vaccination is achieved, the spread of COVID-19 is largely driven by human contact behavior, and I believe this study tries to quantify the effect of public sentiments by incorporating it to the transmission dynamics in a fixed population SIR model. I have several major concerns which may be important to improve the paper.

1. This model assumes that hospital and quarantine does not have a carrying capacity. While violation from hospitals or quarantine may occur, what had been a problem also is the capacity to accommodate such infected individuals once a country exceeds substantial number of cases in facilities. Assuming Vq may partially be one solution for hospitalization, but a more realistic model needs to consider the number of beds available.

2. The model does not consider secondary infection from asymptomatic individuals. Is it true?

3. Once infected, the author assumes lifelong immunity. This should be stated clearly since there are several evidences of reinfection for COVID-19.

4. While in Figure 10, the sentiment-based model seems to overall capture the relative size of the epidemic, the size of the epidemic is generally based on the population size of each country since it is density dependent. Perhaps, in this model, by around day 100, susceptible individuals reach 0 in most countries, while there are still many susceptible individuals in these countries in reality. Also, the cumulative cases in the simulation and reported cases largely differ. Can this be explained by ascertainment of cases?

5. During the process of translation, there are often mistranslation. It is mentioned in line 371-272, a visual inspection was conducted. Were the author’s native speakers in all countries? Even for the same language, perception can differ. For example, British may be more passive aggressive than Americans.

---

## Round 0.2 · Minor Revisions

Please address remaining comments. It must be noted that one of reviewers was not satisfied with your first round revisions. In the next submission, please include point-by-point responses to the first round comments from the reviewer 1, indicating where you have additionally addressed comments in the new round.

Reviewer 1 ·

Basic reporting

Please take the former major comments seriously.

Experimental design

none

Validity of the findings

none

Additional comments

none

Reviewer 2 ·

Basic reporting

The authors improved the model and has clarified most comments. I would like to clarify my comment previously (The second comment on review 1) since the author's were unsure what I meant.

Experimental design

no comment

Validity of the findings

1. Since the effective reproduction number is estimated from the estimated parameters using least square approach, it would be sensible to the estimated parameters. What I meant in the previous review is to compare the estimated effective reproduction number in your model with common methods using epidemic curves:

(1) Jacco Wallinga, Peter Teunis, Different Epidemic Curves for Severe Acute Respiratory Syndrome Reveal Similar Impacts of Control Measures, American Journal of Epidemiology, Volume 160, Issue 6, 15 September 2004, Pages 509–516

(2) Anne Cori, Neil M. Ferguson, Christophe Fraser, Simon Cauchemez, A New Framework and Software to Estimate Time-Varying Reproduction Numbers During Epidemics, American Journal of Epidemiology, Volume 178, Issue 9, 1 November 2013, Pages 1505–1512

2. As for South Africa, the effective reproduction number during lockdown is above unity, thus the number of cases are expected to grow even after the lockdown on 26 March 2020. The validity of this finding may also be verified using the above method (1) or (2) which may be applied easily for example using EpiEstim package in R language.

Additional comments

No comment

---

## Round 0.3 · accepted · Accept

I have read the revised version and the authors' revisions were satisfactory.